# Systematic Review of the Current Status of Human Sarcoma Cell Lines

**DOI:** 10.3390/cells8020157

**Published:** 2019-02-13

**Authors:** Emi Hattori, Rieko Oyama, Tadashi Kondo

**Affiliations:** 1Division of Rare Cancer Research, National Cancer Center Research Institute, Tokyo 1040045, Japan; ehattori@ncc.go.jp; 2Department of Innovative Seeds Evaluation, National Cancer Center Research Institute, Tokyo 1040045, Japan; royama@ncc.go.jp

**Keywords:** sarcoma, patient-derived cancer model, cell line, cell bank, histological subtypes

## Abstract

Sarcomas are rare mesenchymal malignant tumors with unique biological and clinical features. Given their diversity, heterogeneity, complexity, and rarity, the clinical management of sarcomas is quite challenging. Cell lines have been used as indispensable tools for both basic research and pre-clinical studies. However, empirically, sarcoma cell lines are not readily available. To understand the present status of sarcoma cell lines and identify their current challenges, we systematically reviewed reports on sarcoma cell lines. We searched the cell line database, Cellosaurus, and categorized the sarcoma cell lines according to the WHO classification. We identified the number and availability of sarcoma cell lines with a specific histology. We found 844 sarcoma cell lines in the Cellosaurus database, and 819 of them were named according to the WHO classification. Among the 819 cell lines, 36 multiple and nine single cell lines are available for histology. No cell lines were reported for 133 of the histological subtypes. Among the 844 cell lines, 148 are currently available in public cell banks, with 692 already published. We conclude that there needs to be a larger number of cell lines, with various histological subtypes, to better benefit sarcoma research.

## 1. Introduction

Sarcomas are rare mesenchymal malignant tumors with unique biological and clinical features. Sarcomas are unique malignancies for several reasons. First, sarcomas originate from diverse mesenchymal tissue lineages such as adipose, muscle, fibrous, cartilage, nervous, and vascular tissues, or bone. Since these tissues are distributed throughout the human body, sarcomas can occur in almost all organs. Second, sarcomas are heterogeneous diseases that are pathologically grouped into more than 70 described subtypes [1]. The histological appearances do not necessarily represent their normal counterparts, and indeed, the original normal cells are not identified for most sarcomas. Third, sarcomas have high complexity at the molecular level, classifying them into two groups: genetically simple sarcomas, such as those bearing specific genetic alterations, and sarcomas with multiple, complex karyotypic abnormalities with no specific pattern [2,3].

Recent advances in genomic technology using next-generation sequencing have enabled the classification of sarcomas, which did not fit into known specific diagnostic categories [4]. Such classification may lead to innovative therapies. Finally, despite their diversity, heterogeneity, and complexity, sarcomas are rare, accounting for less than 1% of all malignancies. The reasons for this rarity are not well understood. Possible reasons include the need for unique genetic mutations for carcinogenesis, the small number of original cells, and the resistance of the original cells to carcinogenesis. Interestingly, sarcomas are prevalent in children and adolescents, where they account for approximately 20% of cancer-related deaths [5]. Although several molecular predispositions have been suggested [6,7,8,9,10,11,12,13], the mechanisms of early onset of sarcomas in children are not understood.

Given their diversity, heterogeneity, complexity, and rarity, the clinical management of sarcomas is quite challenging. However, there are many successful examples of innovative drugs for sarcomas. For example, a tyrosine kinase inhibitor, imatinib mesylate, was originally developed for the treatment of chronic myelocytic leukemia, which has a unique chromosome translocation; later, imatinib mesylate was repurposed to gastrointestinal stromal tumor (GIST), which is characterized by mutations or overexpression of c-kit and PDGFR [14,15,16,17,18]. Imatinib has also shown activity in metastatic dermatofibrosarcoma protuberans (DFSP) [19] and fibrosarcomatous DFSP [20]. Following imatinib, other tyrosine kinase inhibitors, such as sunitinib [21] and regorafenib [22], have been approved for GISTs.

Other drugs include trabectedin, pazopanib, eribulin, olaratumab, and denosumab. Trabectedin, which binds to the minor groove of DNA to cause DNA damage, demonstrated evidence of cytotoxic activity against soft tissue sarcomas [23,24,25,26,27]. Pazopanib, an inhibitor for VEGFR, PDGFR, FGFR, c-kit, and many other tyrosine kinases [28,29], has suppressive effects on angiogenesis and has been approved for the treatment of non-GIST soft tissue sarcomas [30,31,32]. Eribulin is a microtubule inhibitor, which binds to the vinca domain of tubulin and inhibits the polymerization of tubulin and the assembly of microtubules, inducing cell cycle arrest at the G2/M phase [33,34] and exhibiting anti-tumor effects [35,36]. Eribulin was approved for metastatic breast cancer, and later, it significantly prolonged overall survival in patients with leiomyosarcoma or liposarcoma in a randomized, phase 3 trial with an active control [37]. Olaratumab, a monoclonal antibody targeting platelet-derived growth factor receptor (PDGFR)-alpha, extended the overall survival of metastatic GIST [38] as well as non-GIST sarcomas [39]. New drugs, such as denosumab, a receptor activator of nuclear factor kB ligand (RANKL) [40], have yielded favorable results against osteosarcomas in pre-clinical studies [41,42,43] and giant cell rich tumors [44]. Although the results of these clinical and pre-clinical trials seemed to be encouraging, they were often discrepant and needed to be interpreted with care. For example, Piuperno-Neumann et al. reported a discrepancy between OS206 trial data and preclinical data, and finally did not recommend zoledronate for osteosarcoma patients [45].

This recent progress in the development of novel anti-cancer drugs suggests improvement in the clinical outcomes of patients with sarcomas in the near future [46,47,48]. However, considering the complexity of sarcomas, a larger number of effective anti-cancer drugs should be developed. As the success rate of drug development remains generally low in oncology, and the number of patients who can be recruited into clinical trials is limited for sarcomas, a pre-clinical study to evaluate the eligibility of potential anti-cancer drugs is important, especially in rare malignancies, such as sarcomas.

Cell lines have been used as indispensable tools for both basic research and pre-clinical studies. Since animal cell culture became a common laboratory technique in the mid-1900s, it has served as a model for human cancer research. Since cell lines are maintained in artificial tissue culture conditions, there are critical arguments about the utility of cell lines; however, the advantages of other cancer models, such as xenografts [49] and organoids [50], for cancer research have been emphasized to complement the inherent drawbacks of cell lines. Indeed, the generation of cell lines may involve extensive selection and adaptation to in vitro culture conditions, and in some cell lines, only rare clones may expand with considerable genetic changes [51]. Therefore, the results of experiments using cell lines should be interpreted with caution [52]. However, the advantages of cell lines over other cancer models are obvious. Once the cell lines are stably established, they constantly expand in the tissue culture conditions and they are useful to examine the functional effects and mechanisms of genes or drugs with significant reproducibility. In this context, because cell lines are deposited in public cell banks or shared in the research community, we can integrate research results obtained in different laboratories. Previous reports suggested that the unique characteristics of cell lines can contribute to the development of cancer therapy. For example, cell lines allow the screening of a compound library [53] and the prediction of reactions to treatments [54,55]. In addition to drug development, cell lines are required for biomarker studies. In a biomarker study, the functional evaluation of biomarker candidates is mandatory to convince collaborators to perform multi-institutional validation studies. Cell lines are mandatory to investigate the biological properties of biomarker candidates. Overall, without using cell lines, most of the anti-cancer drugs and biomarkers used in hospitals and the scientific discoveries written in text books could not have been achieved for cancers.

To date, different sarcoma cell lines have been developed. These cell lines represent a useful experimental model to examine the hypothesis about the etiology of diseases, to evaluate the molecular mechanisms of cancer progression, and to examine the effect of potent anti-cancer drugs at the cellular and subcellular levels. At the same time, besides the obvious utilities of cell lines in cancer research, researchers may be empirically aware that sarcoma cell lines are not readily available, probably due to the rarity of the disease, and a lack of proper cell lines hinders basic studies and development of effective therapies for sarcomas. In this review, we provide an overview of the current status of reported sarcoma cell lines, and finally discuss what types of sarcoma cell lines need to be established, what system needs to be created to promote sarcoma research using cell lines, and what biological studies need to be performed to improve the present status of sarcoma cell lines.

## 2. Materials and Methods

### 2.1. Search Strategy

All potentially relevant cell lines were identified by searching the Cellosaurus database (version 28, November 2018) [56,57]. The data file was downloaded from the website of Cellosaurus (https://web.expasy.org/cellosaurus/) and searched using ontology: human cell lines were searched with the term ‘NCBI_TaxID=9606; ‘Homo sapiens’ in ‘Species of origin’, and sarcoma cell lines were searched with a term that was one of the children of the term ‘NCIt:C3810 Connective and Soft Tissue Neoplasm’ in ‘Disease’. The NCI thesaurus ontology file was downloaded from the website of NCI Thesaurus (https://ncit.nci.nih.gov).

### 2.2. Eligibility Criteria and Selection

The following criteria had to be met for a cell line to be included in this review: cell line established from human patients with connective and soft tissue neoplasm, regardless of histology or original sites. Cell lines derived from other cell lines and modified by genes or reagents were considered as duplicates and excluded from the data analysis.

### 2.3. Data Collection Process

The following data were examined for each cell line using Python version 3.6.1 (https://www.python.org/): cell line name, disease, publications, and cell line collections.

### 2.4. Data Items

The focus of this review was on the availability of cell lines with specific sarcoma histology that could influence research activity. Thus, the primary end point of this review was the identification of the histology of the original tumor, as the cell lines are expected to reflect the features of tumors from which they derived; in addition, the study sought to identify what specific histology cell lines are needed to fill in the gaps. The histological classification was performed according to the classification by the World Health Organization [1]. Secondary end points were data availability and publication.

### 2.5. Information Sources

In the Cellosaurus database, the availability of cell lines was examined from the following cell banks: AddexBio, ATCC, BCRC, BCRJ, BEI_Resources, CBA, CCLV, Cell_Biolabs, CLS, Coriell, DGRC, DiscoverX, DSMZ, ECACC, FCDI, ICLC, Imanis, IZSLER, JCRB, KCB, KCLB, KYinno, Millipore, MMRRC, NCBI_Iran, NCI-DTP, NHCDR, NIH-ARP, NISES, RCB (Riken), RSCB, TCB, TKG, TNGB, and WiCell. In addition, we searched the Ximbio website (https://ximbio.com/) for published sarcoma cell lines. The publications in PubMed were also considered for investigation because these cell lines are available from public cell banks or published in academic journals and were, therefore, somehow assured by the public organizations or research community. In addition, the cell lines and their relevant data were supposed to be easily obtainable.

## 3. Results

Cell lines were chosen through a systematic review process (Figure 1). A total of 109,135 cell lines were identified in the Cellosaurus database. Of these, 27,518 cell lines were excluded because they did not originate from humans. In addition, 80,342 cell lines were further excluded because they were not derived from connective and soft tissue neoplasm. Among the resulting 1275 cell lines, 431 cell lines originated from other cell lines and were transfected with genes or treated with reagents; therefore, we excluded these cell lines from the analysis as duplicates. The final number used for the analysis was 844 cell lines; their information was extracted.

### 3.1. Histology

The cell lines were categorized according to the WHO classification. The results are summarized in Appendix A. Among the 189 histological subtypes listed in the WHO classification, 45 had corresponding cell lines, while 133 did not (Appendix A). The histology of original tumors from which the cell lines were most commonly established included Ewing’s sarcoma (156 cell lines), osteosarcoma (148 cell lines), and undifferentiated high-grade pleomorphic sarcoma (43 cell lines). Among the 189 histological subtypes listed in the WHO classification, multiple cell lines were established from 36 histological subtypes of sarcomas. On the other hand, a single cell line was established for each of the following nine histological subtypes: pleomorphic liposarcoma, desmoids-type fibromatosis, tenosynovial giant cell tumor, desmoplastic small round cell tumor, PEComa, osteoblastoma, small cell osteosarcoma, fibrosarcoma of bone, and benign fibrous histiocytoma/non-ossifying fibroma.

We found that there are cell lines that originated from other sarcoma cell lines and were modified by gene transfection or drug treatments (Table 2). Those modified cell lines are most common in osteosarcoma (275 cell lines). Among the 275 cell lines, U2OS cell lines are most commonly used as original cell lines for modifications. We found that there are cell lines for which the reported histology did not match that in the WHO classification (Table 3).

### 3.2. Availability from Cell Banks

Among the 819 cell lines originally derived from patients with connective and soft tissue neoplasm, 139 cell lines were available from the public cell banks (Table 1), while 680 were not. Among the 421 modified cell lines, 262 were available from the public cell banks, and 159 were not (Table 2). In addition to the cell banks examined in Cellosaurus, we searched the Ximbio website for published sarcoma cell lines. By searching the Ximbio website, three cell lines were additionally recognized in the cell bank: 2C4 gamma1A/JAK2 (fibrosarcoma), S_M6R1 (osteosarcoma), and S_N40R2 (osteosarcoma). Among the 35 WHO unclassified cell lines, 12 were available from the public cell banks, and 23 were not (Table 3).

### 3.3. Publication of Cell Lines

Among the 819 cell lines originally derived from patients with connective and soft tissue neoplasm, 674 cell lines were cited in the PubMed database (Table 4), while 145 were not. Among the 421 modified cell lines, 159 were cited in PubMed, and 262 were not (Table 5). Among the 35 WHO unclassified cell lines, 27 were cited in PubMed, and eight were not (Table 6).

### 3.4. Availability and Publication of Cell Lines

The cell lines whose establishment was reported in academic journals that are cited in the PubMed database can be useful research resources because the relevant data of cell lines are available from the published papers. Among the 844 original cell lines, there were 692 cell lines that have been published: among them, 108 cell lines are available from public cell banks (Figure 2A) (Appendix A). Among the 819 original cell lines with the histology defined by WHO classification, there were 674 cell lines that have been published; among them, 103 cell lines are available from public cell banks (Figure 2B) (Appendix A).

## 4. Discussion

A lack of sarcoma cell lines is empirically noticed in the research community; it is important to know their availability from a practical view point. In this review, we investigated the current status of sarcoma cell lines to reveal what cell lines have to be established to promote sarcoma research. The cell line database, Cellosaurus, used in this study includes more than one hundred thousand cell lines and is frequently updated. Thus, Cellosaurus is an adequate cell line database for investigation.

We grouped the cell lines according to the histology of their original tumor. We found that 45 histological subtypes were covered by the currently reported cell lines, while 133 were not. Considering the diversity and complexity of sarcomas, we need more cell lines that represent the different histological subtypes. In addition, we found that multiple cell lines were established for 36 histological subtypes, with a single cell line reported for nine subtypes. During the course of the cell line establishment, clonal selection and expansion may occur, and only limited cell populations may survive under tissue culture conditions. To understand the sustainability of the original characteristics of the established cell lines, the capability of tumor tissue formation and the histology of the formed tumors can be evaluated by xenograft experiments. In addition, patient-to-patient variations are clinically considerable even if they have tumors with the same histology. Therefore, no single cell line can represent the characteristics of whole tumor tissues; we need to use multiple cell lines. In this sense, we also need more cell lines for sarcomas that already have corresponding cell lines.

Cell lines are most frequently established from Ewing’s sarcoma and osteosarcoma samples. However, the absolute number of patients with Ewing’s sarcoma and osteosarcoma is small according to medical statistics; undifferentiated pleomorphic sarcoma, liposarcoma, and leiomyosarcoma are more common sarcomas [58]. Thus, the number of patients may not be a critical factor to determine the established cell lines. Cell lines with a higher malignant potential may be easier to establish, and the clinical stage of donor patients, pathological grading, and prognosis may be correlated with the success rate of the cell line establishment. However, during our investigation, there was no report discussing the efficacy of the cell line establishment in terms of histology. This issue is quite important because we can refine the experimental protocols and improve the efficacy of experiments by clarifying the biological and clinical factors that determine the success rate of establishment.

The histological diagnosis of original tumors of cell lines may need to be updated in cases where the name of cell lines did not match the official classification. Among the 844 sarcoma cell lines investigated, 42 were not named according to the 2013 World Health Organization *Classification of Tumours of Soft Tissue and Bone* [1,59]. The diagnosis of sarcomas has been achieved based on morphological observations, and sarcomas are reclassified by the genetic characterization and subsequent phenotypic correlations. Thus, the diagnosis of cell lines with the official name should be refined by pathological examinations according to the most recent diagnosis criteria. This is a dilemma for a study using clinical materials, because the criteria of histological subtypes may have been updated after the cell lines were reported. To take full advantage of patient-derived sarcoma cell lines, we should investigate the pathology archives and update the diagnosis. However, this will be a challenging task.

Unfortunately, cell lines are not always deposited in cell banks. We found that only 139 of 819 sarcoma cell lines named according to the WHO classification were deposited in public cell banks. Probably, the rest of the cell lines can be provided upon request by researchers. The current cell bank systems may rely on researchers and institutes to undertake the cell line establishment. Establishing novel cell lines costs a considerable amount of resources, such as time and money; furthermore, because cell lines are properties of the institutes to which researchers are affiliated, it may be difficult to deposit all cell lines in public cell banks and share them with other researches. As the establishment of cell lines itself is not necessarily a novel discovery, nor would the publication be in high-impact journals, researchers may not be motivated to establish and share cell lines. A system to motivate cell line establishers and their institutes may be required to improve the availability by depositing cell lines.

This systematic review has several limitations. First, although the genetic background and biological characteristics of some but not all cell lines were reported in publications, this review did not summarize those data. In our research, 692 cell lines were reported in previous papers, and 108 of them were deposited in cell banks (Figure 2). Although the experiments were performed individually using different methods, it is worth integrating the relevant genetic and biological data of reported cell lines to evaluate their possible applications. Second, the clinical features of donor patients, such as metastasis and resistance against therapy, were not investigated in this review. Bernardo et al. [60] performed a systematic review for patient-derived xenografts in bladder cancers and discussed the clinical factors that may influence the take-rate of xenografts. Lu et al. [61] investigated previous studies on xenograft establishment, and correlated the higher engraftment rates with tumor stage. A similar approach could be used for cell lines of sarcomas. Thirdly, the pathological diagnosis should be updated using the most recent pathological criteria of sarcomas. It is possible that some of the reported cell lines may actually represent other subtypes. However, because we cannot access the original pathological archives and it takes too much effort to validate the results of pathological diagnosis, we cannot know the correct histology according to the most recent WHO classification. This is a general problem of sarcoma research, as observed when we conducted histology-based research using previously published data. Finally, the applications of cell lines are diverse, and probably depend on the cell lines and the experiments. In addition to the number of established cell lines, it would be worth investigating the literature to determine how the established cell lines were used by the researchers who received them.

## 5. Conclusions

Cell lines have been considered a valuable tool for both basic research and pre-clinical studies. The functional significance of genetic products such as mRNA, miRNA, and proteins can be clarified using living cells, and cell lines are an indispensable research resource. In the preclinical evaluation of new drugs, their tumor suppressive effects and mode-of-action are also investigated using cell lines. Although the predictive power of cell lines can be undermined by the selective pressures during the process of establishment and long-term passaging, a great advantage of cell lines is that the examinations can be done in a high-throughput manner with relatively low costs. Patient-derived xenografts (PDXs) may complement the inherent drawbacks of cell lines, because PDXs may retain the microenvironmental conditions of the original tumors. However, the manipulation of PDX requires time-consuming and tremendous efforts, and their unstable molecular backgrounds have been revealed at the genome level. Moreover, because the human stromal components in PDX tumors are replaced with mouse ones after several passes, consistent results may be limited in experiments using PDXs. Taken together, cell lines have a unique utility, and they are indispensable in cancer research.

We conclude that (1) more sarcoma cell lines representing the various histological types as well as those established by single cell lines are needed to effectively capture the diversity and complexity of the disease; (2) a system is needed to reward the efforts of researchers who establish and deposit cell lines into public cell banks to promote cell line sharing in the research community, and (3) further investigations are required to determine the critical factors determining the success rate of the cell line establishment and create effective experimental protocols.

## Figures and Tables

**Figure 1 cells-08-00157-f001:**
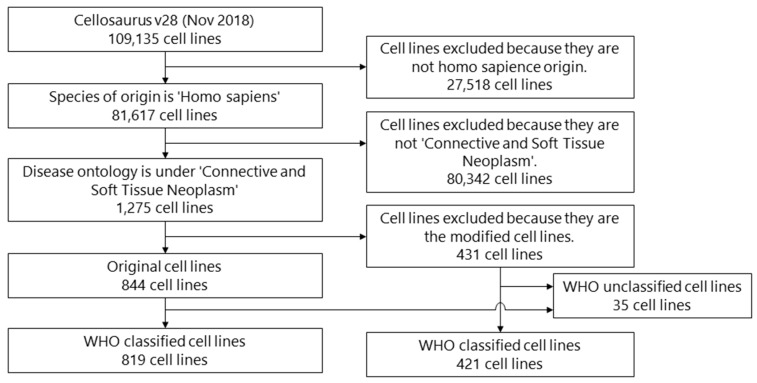
Flow diagram of the systematic review process.

**Figure 2 cells-08-00157-f002:**
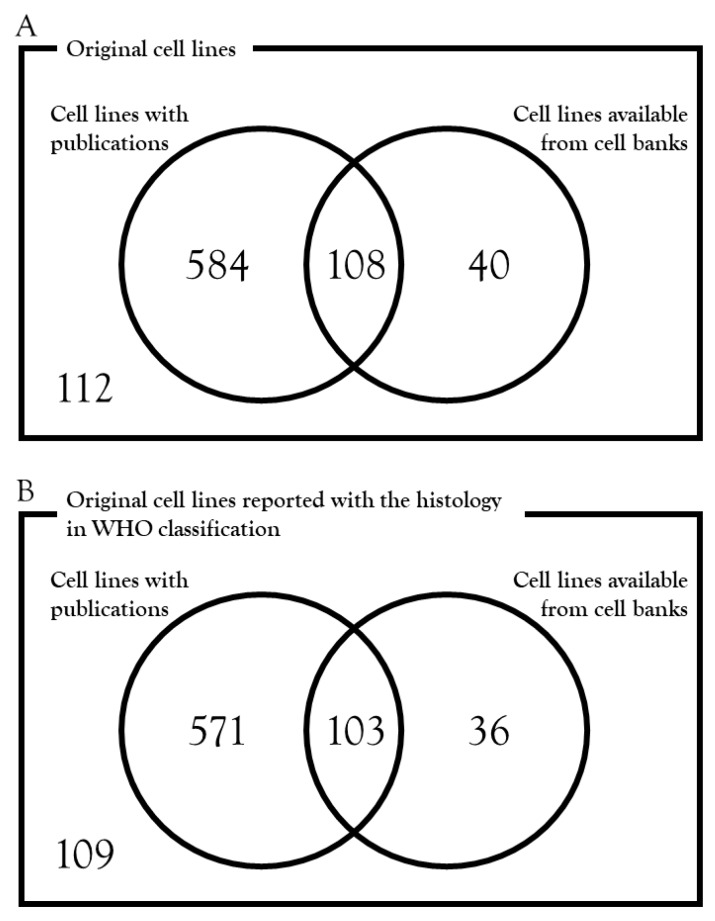
The number of cell lines that were reported in publications or available from cell banks. (**A**) Note that among the 844 original cell lines, 112 cell lines were neither published nor available from cell banks. (**B**) Note that among the 819 original cell lines with the histology defined by WHO classification, 109 cell lines were neither published nor available from cell banks.

**Table 1 cells-08-00157-t001:** Availability of WHO classified original cell lines.

Group	Disease	In Public Cell Banks	Not in Public Cell Banks	Total
Adipocytic tumors	Lipoma	0	6	6
	Dedifferentiated liposarcoma	2	40	42
	Myxoid liposarcoma	0	9	9
	Pleomorphic liposarcoma	0	1	1
Fibroblastic/myofibroblastic tumors	Desmoids-type fibromatosis	1	0	1
	Dermatofibrosarcoma protuberans	2	2	4
	Myxofibrosarcoma	2	5	7
	Fibrosarcoma	9	21	30
So-called fibrohistiocytic tumors	Tenosynovial giant cell tumor	0	1	1
Smooth-muscle tumors	Leiomyoma of deep soft tissue	0	10	10
	Leiomyosarcoma	6	10	16
Pericytic (perivascular) tumors	Glomus tumors	1	2	3
Skeletal-muscle tumors	Embryonal rhabdomyosarcoma	6	31	37
	Alveolar rhabdomyosarcoma	6	40	46
	Pleomorphic rhabdomyosarcoma	0	2	2
	Rhabdomyosarcoma	5	14	19
Vascular tumors	Lymphangioma	2	0	2
	Kaposi sarcoma	0	8	8
Gastrointestinal stromal tumors	Gastrointestinal stromal tumors	2	8	10
Nerve sheath tumors	Malignant peripheral nerve sheath tumor	5	19	24
Tumors of uncertain differentiation	Myoepithelioma/myoepithelial carcinoma/mixed tumor	0	2	2
	Synovial sarcoma	5	29	34
	Epithelioid sarcoma	4	12	16
	Alveolar soft-part sarcoma	1	1	2
	Clear cell sarcoma of soft tissue	1	12	13
	Desmoplastic small round cell tumor	0	1	1
	Extrarenal rhabdoid tumor	0	12	12
	PEComa	0	1	1
	Intimal sarcoma	1	3	4
Chondrogenic tumors	Chondrosarcoma	5	34	39
	Dedifferentiated chondrosarcoma	0	5	5
Osteogenic tumors	Osteoid osteoma	2	1	3
	Osteoblastoma	0	1	1
	Small cell osteosarcoma	0	1	1
	Osteosarcoma	37	111	148
Fibrogenic tumors	Fibrosarcoma of bone	0	1	1
Fibrohistiocytic tumors	Benign fibrous histiocytoma/non-ossifying fibroma	0	1	1
Ewing sarcoma	Ewing sarcoma	14	142	156
Osteoclastic giant cell rich tumors	Giant cell tumor of bone	4	4	8
Notochordal tumors	Chordoma	7	16	23
Vascular tumors	Angiosarcoma	2	2	4
Myogenic, lipogenic and epithelial tumors	Bone leiomyosarcoma	0	2	2
	Liposarcoma	2	16	18
Undifferentiated high-grade pleomorphic sarcoma	Undifferentiated high-grade pleomorphic sarcoma	4	39	43
Tumor syndromes	Multiple osteochondromas	1	2	3
	**Total**	139	680	819

**Table 2 cells-08-00157-t002:** Availability of WHO classified and modified cell lines.

Group	Disease	In Public Cell Banks	Not in Public Cell Banks	Total
Adipocytic tumors	Dedifferentiated liposarcoma	0	1	1
	Myxoid liposarcoma	0	1	1
Fibroblastic/myofibroblastic tumors	Fibrosarcoma	26	17	43
Smooth-muscle tumors	Leiomyosarcoma	1	10	11
Skeletal-muscle tumors	Embryonal rhabdomyosarcoma	2	17	19
	Alveolar rhabdomyosarcoma	0	23	23
Gastrointestinal stromal tumors	Gastrointestinal stromal tumors	0	7	7
Nerve sheath tumors	Malignant peripheral nerve sheath tumor	0	1	1
Tumors of uncertain differentiation	Epithelioid sarcoma	0	5	5
	Intimal sarcoma	2	5	7
Chondrogenic tumors	Chondrosarcoma	0	6	6
Osteogenic tumors	Osteosarcoma	228	47	275
Ewing sarcoma	Ewing sarcoma	1	11	12
Notochordal tumors	Chordoma	0	1	1
Vascular tumors	Angiosarcoma	2	2	4
Undifferentiated high-grade pleomorphic sarcoma	Undifferentiated high-grade pleomorphic sarcoma	0	5	5
	**Total**	262	159	421

**Table 3 cells-08-00157-t003:** Availability of cell lines with histology unclassified by WHO.

Group	Disease	In Public Cell Banks	Not in Public Cell Banks	Total
Not_classified	Ovarian mixed germ cell tumor	2	5	7
	Thyroid gland sarcoma	2	0	2
	Endometrioid stromal sarcoma	2	3	5
	Soft tissue sarcoma	0	2	2
	Sarcoma	0	8	8
	Histiocytoma	1	0	1
	Skin sarcoma	2	0	2
	Uterine corpus sarcoma	3	3	6
	Meningeal sarcoma	0	1	1
	Benign synovial neoplasm	0	1	1
	**Total**	12	23	35

**Table 4 cells-08-00157-t004:** PubMed citation of WHO classified original cell lines.

Group	Disease	PubMed Cited	Not Cited	Total
Adipocytic tumors	Lipoma	5	1	6
	Dedifferentiated liposarcoma	41	1	42
	Myxoid liposarcoma	9	0	9
	Pleomorphic liposarcoma	1	0	1
Fibroblastic/myofibroblastic tumors	Desmoids-type fibromatosis	0	1	1
	Dermatofibrosarcoma protuberans	3	1	4
	Myxofibrosarcoma	6	1	7
	Fibrosarcoma	12	18	30
So-called fibrohistiocytic tumors	Tenosynovial giant cell tumor	1	0	1
Smooth-muscle tumors	Leiomyoma of deep soft tissue	9	1	10
	Leiomyosarcoma	11	5	16
Pericytic(perivascular) tumors	Glomus tumors	0	3	3
Skeletal-muscle tumors	Embryonal rhabdomyosarcoma	36	1	37
	Alveolar rhabdomyosarcoma	45	1	46
	Pleomorphic rhabdomyosarcoma	2	0	2
	Rhabdomyosarcoma	13	6	19
Vascular tumors	Lymphangioma	0	2	2
	Kaposi sarcoma	7	1	8
Gastrointestinal stromal tumors	Gastrointestinal stromal tumors	10	0	10
Nerve sheath tumors	Malignant peripheral nerve sheath tumor	22	2	24
Tumors of uncertain differentiation	Myoepithelioma/myoepithelial carcinoma/mixed tumor	2	0	2
	Synovial sarcoma	31	3	34
	Epithelioid sarcoma	15	1	16
	Alveolar soft-part sarcoma	2	0	2
	Clear cell sarcoma of soft tissue	13	0	13
	Desmoplastic small round cell tumor	1	0	1
	Extrarenal rhabdoid tumor	11	1	12
	PEComa	0	1	1
	Intimal sarcoma	4	0	4
Chondrogenic tumors	Chondrosarcoma	34	5	39
	Dedifferentiated chondrosarcoma	5	0	5
Osteogenic tumors	Osteoid osteoma	0	3	3
	Osteoblastoma	0	1	1
	Small cell osteosarcoma	1	0	1
	Osteosarcoma	100	48	148
Fibrogenic tumors	Fibrosarcoma of bone	1	0	1
Fibrohistiocytic tumors	Benign fibrous histiocytoma/non-ossifying fibroma	1	0	1
Ewing sarcoma	Ewing sarcoma	136	20	156
Osteoclastic giant cell rich tumors	Giant cell tumor of bone	5	3	8
Notochordal tumors	Chordoma	20	3	23
Vascular tumors	Angiosarcoma	4	0	4
Myogenic, lipogenic and epithelial tumors	Bone leiomyosarcoma	2	0	2
	Liposarcoma	10	8	18
Undifferentiated high-grade pleomorphic sarcoma	Undifferentiated high-grade pleomorphic sarcoma	42	1	43
Tumor syndromes	Multiple osteochondromas	1	2	3
	**Total**	674	145	819

**Table 5 cells-08-00157-t005:** PubMed citation of WHO classified and modified cell lines.

Group	Disease	PubMed Cited	Not Cited	Total
Adipocytic tumors	Dedifferentiated liposarcoma	1	0	1
	Myxoid liposarcoma	1	0	1
Fibroblastic/myofibroblastic tumors	Fibrosarcoma	18	25	43
Smooth-muscle tumors	Leiomyosarcoma	11	0	11
Skeletal-muscle tumors	Embryonal rhabdomyosarcoma	18	1	19
	Alveolar rhabdomyosarcoma	20	3	23
Gastrointestinal stromal tumors	Gastrointestinal stromal tumors	7	0	7
Nerve sheath tumors	Malignant peripheral nerve sheath tumor	1	0	1
Tumors of uncertain differentiation	Epithelioid sarcoma	5	0	5
	Intimal sarcoma	3	4	7
Chondrogenic tumors	Chondrosarcoma	6	0	6
Osteogenic tumors	Osteosarcoma	51	224	275
Ewing sarcoma	Ewing sarcoma	9	3	12
Notochordal tumors	Chordoma	1	0	1
Vascular tumors	Angiosarcoma	2	2	4
Undifferentiated high-grade pleomorphic sarcoma	Undifferentiated high-grade pleomorphic sarcoma	5	0	5
	**Total**	159	262	421

**Table 6 cells-08-00157-t006:** PubMed citation of cell lines with histology unclassified by WHO.

Group	Disease	PubMed Cited	Not Cited	Total
Not_classified	Ovarian mixed germ cell tumor	6	1	7
	Thyroid gland sarcoma	1	1	2
	Endometrioid stromal sarcoma	5	0	5
	Soft tissue sarcoma	2	0	2
	Sarcoma	6	2	8
	Histiocytoma	0	1	1
	Skin sarcoma	0	2	2
	Uterine corpus sarcoma	5	1	6
	Meningeal sarcoma	1	0	1
	Benign synovial neoplasm	1	0	1
	**Total**	27	8	35

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
