# Peer review of "Systematic Review of the Current Status of Human Sarcoma Cell Lines"

_cells, 2019, doi:10.3390/cells8020157_

Round 1

Reviewer 1 Report

The manuscript « Systematic Review of Current Status of Human Sarcoma Cell Lines» analyses the availability of sarcoma cell lines in comparison to the diversity of sarcomas following the WHO classification. Challenges of such study are well-introduced but few supplementary comments may be needed.

In the introduction, the listing of mesenchymal tissue lineages should be completed in regard to the tumour groups within tables. Indeed adipose, muscle, fibrous, cartilage and bone were listed (line 29), while nervus and vascular tissues were missing.

Denosumab, which targets osteoclast differentiation, was introduced by authors as a new drug with encouraging results against osteosarcomas in pre-clinical studies (lines 67-69). However such encouraging preclinical results have to be commented carefully in regard to clinical results obtained with Zoledronate, another osteoclast inhibitor, in combination with chemotherapy and surgery to treat osteosarcoma. Piperno-Neumann S et al. (Lancet Oncol. 2016 Aug;17(8):1070-1080) reported discrepancy between OS2006 trial and preclinical data and finally do not recommend zoledronate in osteosarcoma patients. Additionally, authors may comment the use of Denosumab in osteoclastic giant cell rich tumours when they are not resectable.

The sentence (line 169) “We found that there are cell lines the histology of which 169 corresponded to original tumors did not match to the WHO classification” has to be reformulated. Does it indicate that the histology of the original tumour which was reported with a cell line does not match with the WHO classification?

Histological subtype correspondences between cell-line-induced tumours in animals and original tumours in patients may be of interest. In the discussion, authors may comment that histological data concerning tumours that can be induced with each cell line in small animals may be carefully analysed in comparison to original patient tumour. Indeed, it may be important to repeat that one work hypothesis of the authors was that “cell lines are expected to reflect the features of tumors from which they derived.” (line 131). However, MG63, U2OS and modified-HOS cell lines, which are commonly used to mimic osteosarcoma are poorly osteogenic by themselves in mice, in contrast to human osteosarcoma. They may induce bone remodeling, neoformation associated with osteolysis, but they produced undifferentiated tissues with poor evidence of osteoid matrice secreted by tumour cells. Authors may choose different tumour examples to illustrate such limit and to complete their paragraph between lines 228 to 238.

Line 209, authors may refer to supplementary table 2 at the sentence end.

A short legend should be added to each table.

Authors have used both “tumors” and “tumours”; they may replace “tumors” by “tumours.

Author Response

Reviewer #1

Comment 1

In the introduction, the listing of mesenchymal tissue lineages should be completed in regard to tumour groups within tables. Indeed adipose, muscle, fibrous, cartilage and bone were listed (line 29), while nervus and vascular tissues were missing.

Response 1

As suggested, we have completed the list of normal counterpart tissues in the revised manuscript.

Comment 2

Denosumab, which targets osteoclast differentiation, was introduced by authors as a new drug with encouraging results against osteosarcomas in pre-clinical studies (lines 67-69). However such encouraging preclinical results have to be commented carefully in regard to clinical results obtained with Zoledronate, another osteoclast inhibitor, in combination with chemotherapy and surgery to treat osteosarcoma. Piperno-Neumann S et al. (Lancet Oncol. 2016 Aug;17(8):1070-1080) reported discrepancy between OS2006 trial and preclinical data and finally do not recommend zoledronate in osteosarcoma patients. Additionally, authors may comment the use of Denosumab in osteoclastic giant cell rich tumours when they are not resectable.

Response 2

Accordingly, we have cited the recommended paper about Zoledronate and discussed its possible utility for osteoclastic giant cell rich tumours.

Comment 3

The sentence (line 169) We found that there are cell lines the histology of which 169 corresponded to original tumors did not match to the WHO classificationhas to be reformulated. Does it indicate that the histology of the original tumour which was reported with a cell line does not match with the WHO classification?

Response 3

The reported histology of the cell lines did not match that in the WHO classification. We have clarified this point in the revised manuscript.

Comment 4

Histological subtype correspondences between cell-line-induced tumours in animals and original tumours in patients may be of interest. In the discussion, authors may comment that histological data concerning tumours that can be induced with each cell line in small animals may be carefully analysed in comparison to original patient tumour. Indeed, it may be important to repeat that one work hypothesis of the authors was that cell lines are expected to reflect the features of tumors from which they derived.(line 131). However, MG63, U2OS and modified-HOS cell lines, which are commonly used to mimic osteosarcoma are poorly osteogenic by themselves in mice, in contrast to human osteosarcoma. They may induce bone remodeling, neoformation associated with osteolysis, but they produced undifferentiated tissues with poor evidence of osteoid matrice secreted by tumour cells. Authors may choose different tumour examples to illustrate such limit and to complete their paragraph between lines 228 to 238.

Response 4

We totally agree with the reviewer’s comment. We never meant that cell lines perfectly reflect the phenotypes of original tumours. Histology of tumours formed by cell lines may be good indicators of the sustainability of the original characteristics tumours. We have clarified this point in the revised manuscript.

Comment 5

Line 209, authors may refer to supplementary table 2 at the sentence end.

Response 5

It would not be appropriate to refer to Supplementary Table 2 at the indicated section as we did not mention the WHO classification in this section. Instead, we have referred to Supplementary Table 1 wherein all data for Figure 2 have been summarized in the revised manuscript.

Comment 6

A short legend should be added to each table.

Response 6

We do not think that explanations are needed in the present tables. The titles of the tables explain the contents.

Comment 7

Authors have used both tumors and tumours; they may replace tumorsby tumour.

Response 7

Accordingly, we have done so in the revised manuscript.

Reviewer 2 Report

This manuscript is a very useful inventorization of cell lines that may be helpful to the sarcoma research community. It shows an analysis of all sarcoma cell lines that are collected in the Cellosaurus database. Herewith some suggestions to improve the manuscript

The paper heavily relies on the Cellosaurus database, which obviously has its limitations, such as: not ALL cell lines ever reported are included, diagnosis of the presumed cell line is uncertain, since there is no quality check on the primary tissue from which the cells were derived, mis-identification is not excluded etc. All these laws are copied to this study, so the authors should add this disclaimer in their manuscript.

It is not clear which information has been generated by the authors and which has been extracted directly from the Cellosaurus database. E.g., according to line 136 on page 3 it seems that the authors have examined all the cell banks mentioned here, but this information is generated by the Cellosaurus team.

It is very much appreciated that the authors show their concern with WHO nomenclature and that non-WHO diagnoses of cell lines are separated from the remaining lines. However, in Table 2 there are still non-WHO entries, like fibrosarcoma and glomus-tumor. It is obvious that not all entries are in accordance with WHO nomenclature since they are are entered in a database that gets data from all different sources and different time periods, but the

authors pretend that they have cleaned these data and that is not correct.

Figure 1, the second panel on the right side is not correct, it should state ‘not connective and soft tissue neoplasms’

The headings in the tables ‘available’ and ‘not available’ are misleading. It would be more correct to call them ‘in public cell banks’ and ‘not in public cell banks’

Please clarify how a single cell line is reported for 9 subtypes. Which cell line is this and how can this happen?

The pPNET can be reclassified as Ewing sarcoma

Author Response

Reviewer#2 Comment 1 The paper heavily relies on the Cellosaurus database, which obviously has its limitations, such as: not ALL cell lines ever reported are included, diagnosis of the presumed cell line is uncertain, since there is no quality check on the primary tissue from which the cells were derived, mis-identification is not excluded etc. All these laws are copied to this study, so the authors should add this disclaimer in their manuscript. Response 1 Firstly, we need to rely on a database, and we used Cellosaurus. We have discussed the reason why Cellosaurus is an adequate database for our review in the original manuscript. The cell line database, Cellosaurus, used in this study includes more than hundred thousand cell lines and is frequently updated. Thus, Cellosaurus is an adequate cell line database for investigation. Secondly, the pathological diagnosis of original tumour tissues cannot be validated after publication. This is not a problem unique to our review. The criteria for the pathological diagnosis have been updated, and because the original pathological archives are not opened to the research community, we cannot diagnose the samples again. We discussed this issue in the original manuscript. The histological diagnosis of original tumours of cell lines may need to be updated in cases where the name of cell lines did not match the official classification. Among the 844 sarcoma cell lines investigated, 42 were not named according to the 2013 World Health Organization Classification of Tumours of Soft Tissue and Bone [1, 59]. Diagnosis of sarcomas has been achieved based on morphological observations, and sarcomas are reclassified by the genetic characterization and subsequent phenotypic correlations. Thus, diagnosis of cell lines with the official name should be refined by pathological examinations according to the most recent diagnosis criteria. This is a dilemma for a study using clinical materials because the criteria of histological subtypes may have been updated after the cell lines were reported. To take full advantage of patient-derived sarcoma cell lines, we should investigate the pathology archives and update the diagnosis. However, this will be a challenging task. Comment 2 It is not clear which information has been generated by the authors and which has been extracted directly from the Cellosaurus database. E.g., according to line 136 on page 3 it seems that the authors have examined all the cell banks mentioned here, but this information is generated by the Cellosaurus team. Response 2 In addition to the cell banks examined in the Cellosaurus database, we searched the Ximbio for the sarcoma cell lines, and added three new cell lines in the cell banks. We have clarified this point in the revised manuscript. Comment 3 It is very much appreciated that the authors show their concern with WHO nomenclature and that non-WHO diagnoses of cell lines are separated from the remaining lines. However, in Table 2 there are still non-WHO entries, like fibrosarcoma and glomus-tumor. It is obvious that not all entries are in accordance with WHO nomenclature since they are entered in a database that gets data from all different sources and different time periods, but the authors pretend that they have cleaned these data and that is not correct. Response 3 We examined all the histological names of the cell lines and selected the histology described in the WHO classification. Fibrosarcoma of bone and glomus-tumour are included in the WHO classification. Reference Fletcher CDM, Bridge JA, Hogendoorn P, Mertens F. WHO Classification of Tumours of Soft Tissue and Bone. Fourth Edition edn. Geneva: WHO Press, 2013. Page 91 fibrosarcoma Page 116 glomus-tumour Comment 4 Figure 1, the second panel on the right side is not correct, it should state ”not connective and soft tissue neoplasms” Response 4 Accordingly, we have corrected Figure 1 in the revised manuscript. Comment 5 The headings in the tables ”available”and ”not available”are misleading. It would be more correct to call them ”in public cell banks”and “not in public cell banks” Response 5 Accordingly, we have revised the tables. Comment 6 Please clarify how a single cell line is reported for 9 subtypes. Which cell line is this and how can this happen? Response 6 A single cell line is available for each of the following nine histological subtypes: pleomorphic liposarcoma, desmoids-type fibromatosis, tenosynovial giant cell tumour, desmoplastic small round cell tumour, PEComa, osteoblastoma, small cell osteosarcoma, fibrosarcoma of bone, and benign fibrous histiocytoma/non-ossifying fibroma. We have clarified this point in the revised manuscript. Comment 7 The pPNET can be reclassified as Ewing sarcoma Response 7 We have included pPNET in the classification of Ewing’s sarcoma and have accordingly revised Figures 1 and 2, and Tables 1, 2, 3, 4, 5, and 6, and Supplementary Table 1.